# Does the Gut Microbial Metabolome Really Matter? The Connection between GUT Metabolome and Neurological Disorders

**DOI:** 10.3390/nu14193967

**Published:** 2022-09-24

**Authors:** Małgorzata Anna Marć, Rafał Jastrząb, Jennifer Mytych

**Affiliations:** Research and Development Center, Olimp Laboratories Sp. z o.o, Pustynia 84F, 39-200 Dębica, Poland

**Keywords:** microbiome, metabolome, probiotics, neurodegeneration, Parkinson’s disease, Alzheimer’s disease, gut microbiota

## Abstract

Herein we gathered updated knowledge regarding the alterations of gut microbiota (dysbiosis) and its correlation with human neurodegenerative and brain-related diseases, e.g., Alzheimer’s and Parkinson’s. This review underlines the importance of gut-derived metabolites and gut metabolic status as the main players in gut-brain crosstalk and their implications on the severity of neural conditions. Scientific evidence indicates that the administration of probiotic bacteria exerts beneficial and protective effects as reduced systemic inflammation, neuroinflammation, and inhibited neurodegeneration. The experimental results performed on animals, but also human clinical trials, show the importance of designing a novel microbiota-based probiotic dietary supplementation with the aim to prevent or ease the symptoms of Alzheimer’s and Parkinson’s diseases or other forms of dementia or neurodegeneration.

## 1. Introduction

The gut microbial community is the largest micro-ecosystem in the human body, formed by more than 1000 different species, with high density per mL and encoding approximately 4 × 10^6^ genes, 150 times more unique than the human genome [1]. It is predicted that the total weight of our gut microbiota deposit circulates around 1–2 kg, approximately reaching the human brain weight [2]. Ninety-five percent of symbiotic microorganisms are located in the gut [3,4]. These multi-species ecological arrangements mostly comprise anaerobic bacteria, with at least 500–1000 different species and more than 7000 strains, mainly from the *Firmicutes* and *Bacteroidetes* families (accounting for approx. 90% of bacterial species). In addition, the gut microbiome is co-inhabited by bacteria belonging to *Proteobacteria*, *Actinobacteria*, and *Verrucomicrobia* phyla, as well as archaea, viruses, fungi, bacteriophages, and protozoa (especially during pathologic conditions), which reside in the gastrointestinal (GI) tract, maintaining symbiosis with the host [5,6,7].

The gut microbiome is crucially engaged in the regulation and maintenance of human health and homeostasis, i.e., proper intestine motion, fibre digestion, regulating the development of host immunity, and exerting a protective effect against pathogenic factors (such as *Campylobacter jejuni*, *Helicobacter pylori*, and *Clostridium difficile*) [8,9,10]. The Irish ELDERMET study (Elderly to benefit from Research Project) demonstrated the correlation between intestinal microbiota diversity and health outcomes such as overall health, frailty, and proper immune functions, underlying the gut microbiota composition as a possible indicator of healthy ageing [11]. Multiple studies showed the interplay between the intestinal microbiota and modulation of central nervous system (CNS) functions and neurodegeneration processes [11], pointing to the microbiome-gut–brain axis as one of the factors modulating host homeostasis in health and disease. In the past decade, the alterations of the gut microbiota, which are called dysbiosis, have been connected with the pathophysiology of numerous common diseases such as type 2 diabetes, obesity, gastrointestinal diseases, and cardiovascular events. Dysbiosis state is also implicated in neurological and neurodegenerative disorders such as autism, and Alzheimer’s (AD) or Parkinson’s (PD) diseases [9,12]. During dysbiosis, disrupted homeostasis allows the pathobionts to expand, which can further amend the bowel’s mucosal barrier, exerting neurogenic inflammatory processes [9,13,14].

However host–microbiota interplay is not exclusively restricted to cell–cell interactions. Intestinal microbes are capable of producing a plethora of different types of compounds, i.e., small molecules (amino acids, neurotransmitters, short-chain fatty acids (SCFAs) and other fatty acid derivatives, indoles, vitamins, and cofactors), polysaccharides, and proteins (i.e., bacteriocins), which have been reported as important factors in shaping the bacterial community and modulation of the immune system [15]. It has been demonstrated that intestinal bacterial strains such as *Escherichia coli* or *Lactobacillus* spp. interact directly with the host’s CNS (central nervous system), as they can release neurotransmitters such as dopamine, noradrenaline, histamine, acetylcholine, GABA (gamma-aminobutyric acid), or serotonin [16]. This metabolic status of the microbiome is also vulnerable to dysbiosis and compositional changes and often leads to metabolic disturbances. For example, dysbiosis can induce elevated reactive oxygen species (ROS) levels, intensifying oxidative stress and neuronal inflammatory processes [10,17]. It has been estimated that 40% of the metabolites detected in humans are the products of bacterial metabolism [18], which plays a fundamental role in the modulation of the host’s physiological processes, including immune-mediated neurodegeneration or affecting neural cell metabolism [19].

In this review, we summarized the influence of microbial metabolites on the severity and progression of neurodegenerative diseases and their correlation with microbial AD/PD-associated dysbiosis, pointing out particular genera/phyla responsible for increased/decreased production. Finally, the review concludes with a brief summary of actual knowledge regarding possible interventions to reverse the negative effects of a disbalanced microbiome in neurodegenerative diseases.

## 2. Microbiota–Brain–Gut Axis and Neurodegenerative Diseases/Brain Disorders

Some research reports underline the crucial role of the microbiome in the normal development, regulation of behavior, cognition, and brain functions, but also in the etiology, the pathogenesis, and the progression of debilitating neurological disorders, e.g., Alzheimer’s and Parkinson’s diseases, autism spectrum disorder, multiple sclerosis, or even bipolar disorder, depression, schizophrenia, Huntington’s, stroke, and brain ageing [18,20]. Animal studies have demonstrated the participation of microbiota in neural-related processes such as neurodevelopment, neuroinflammation, interactions with neurons and glia, and behavior [11]. It has been postulated that microbiome influences neurological homeostasis via the microbiome–gut–brain axis.

The gut microbiota and brain crosstalk is a complex process including the vagus nerve, the immune system, gut hormone signaling pathways, tryptophan amino acid metabolism, hypothalamic–pituitary–adrenal axis, and the products of bacterial metabolism such as short-chain fatty acids [21] or gut-derived neurotransmitters [22] (Figure 1). Experimental studies demonstrated that even minor modifications in gut microbiota composition can lead to serious modification of brain functions, subsequently affecting intestinal activity through the secretion of specific hormones, neuropeptides, and neurotransmitters [23]. For example, butyrate, acetate, or propionate can induce the release of leptin and glucagon-like peptide-1, which are the gut hormones interacting with the vagus nerve and brain receptors [24,25,26]. In a mice model, the reduction of the variety of microbiota significantly diminished the blood–brain barrier (BBB) expression of tight junction proteins, occludin and claudin-5, disrupting the barrier function of endothelial cells. Therefore, rodent neural cells become more susceptible to stimuli produced by bacteria, among them lipopolysaccharide (LPS) and factors such as oxidative stress mediators [27]. Moreover, defects in mucosal barrier tight junction proteins increase intestinal permeability, causing the “leaky gut” effect followed by systemic inflammation and pathogenic agents translocation into the bloodstream (e.g., LPS) with elicit pro-inflammatory cytokine release [28,29]. Aoyama et al. provided evidence of the spontaneous neutrophil apoptosis triggered by butyrate and propionate through a histone deacetylase (HDAC) inhibition, the expression of caspase-3, caspase-8, and caspase-9 pathways, and proteins belonging to the Bcl-2 family, affecting the intestines and different organs. Butyrate and propionate activated neutrophil apoptosis both in the absence and presence of LPS or tumor necrosis factor α (TNF-α), and therefore in normal and pro-inflammatory conditions. Moreover, the expression of the proapoptotic *bax* mRNA was significantly elevated and, just the opposite, the expression level of antiapoptotic *mcl-1* and *a1* mRNA was importantly decreased by the abovementioned SCFAs in non-activated neutrophils. LPS induction highly increased the expression of *mcl-1* and *a1* mRNA; interestingly, this effect was neutralized by SCFAs [30]. Moreover, clinical trials suggest an altered gut microbial composition of patients with neurodegenerative disorders, together with significant differences in microbial and serum metabolomic profiles [31]. These data indicate that both microbiome composition and microbiome metabolome affect neurological diseases. It is worth mentioning that gut peripheral tissues are a residue for 70% of the human immune system, which is a well-known mediator of changes in gut ecosystem, inflammatory response, and also regulates multiple processes in CNS [32]. A good example of the interplay between gut-located immune cells and brain cells is the influence of regulatory T cells, which exert a neuroprotective effect by stimulating oligodendrocyte differentiation and triggering neuronal remyelination by interleukin IL-10 [19]. At the molecular level, there is evidence that mammalian Toll-Like Receptors (TLRs) play an important role in neurodegeneration. TLRs receptors are one of the main receptor families engaged in transducing and shaping innate immune responses. They are expressed both in immune and non-immune cells, especially increasing after contact with microbial pathogens or bacterial elements, i.e., cell walls, peptidoglycans, or DNA [33]. TLR activation (mainly TLR2 and TLR4) induces an inflammatory response cascade, preceding the neuronal loss characteristic for Parkinson’s disease [34], which could be a result of a “cascade reaction”: disrupted microbial composition and metabolome, increased permeability of the gut cell wall, and increased exposure to TLR ligands. Moreover, the microbiome and its metabolites are associated with immune-mediated neuroinflammation processes, brain injury, and neurogenesis [35]. Microglia cells are brain glial-resident immune cells responsible for maintaining proper immune functions including phagocytosis, antigen presentation, the production of cytokines, and the subsequent inflammatory reactions [36,37]. Research reports indicate the influence of microbiota on microglial maturation and function; however, this mechanism is still unclear [38]. It has been confirmed that gut dysbiosis can augment intestinal permeability and bacterial translocation, causing an over-response of the immune system and the subsequent systemic/central nervous system inflammation [39,40]. The adaptive immune system also participates in the regulation of healthy microbiota. The crucial players in intestinal homeostasis maintenance are B cells, which produce secretory IgA antibodies targeted toward specific bacteria [41]. Additionally, gut microbiota can stably influence the host’s gene expression through epigenetic mechanisms, including histone modifications, methylation of DNA, and non-coding RNA expression [42]. Overall, the microbiome plays a big role in moderating the maturation of immune cells residing in the CNS tissues via an interplay and crosstalk with the peripheral immune system. Gut microbiota act as a multifunctional hub modulating the immune system via the production of immunomodulatory and anti-inflammatory signaling molecules, reaching immune cells. Commensal microbes fabricate various metabolites from digested food, SCFAs among them, which participate in the maintenance of intestinal homeostasis, exerting anti-inflammatory activity on the intestinal mucosa [43]. The gut microbiota participates in signal transduction pathways as it can communicate between the host’s innate immune system cells, which are located at the interface between the host and the microbiome [44].

## 3. Microbiota Metabolites and the Severity of Neurodegenerative Diseases

Scientific evidence indicates that the metabolic status of microbiota exerts a more important role than the equilibrium between bacterial species. Microbial metabolites can act as positive modulators of the gut–brain-axis constituents, boosting the immune cells and exerting a positive/protective effect on neurodegenerative disease progression [45]. On the other hand, disrupted equilibrium between microbial species often leads to specific switches in the microbial metabolome, which may be harmful and associated with neurodegenerative disease progression. For example, dysregulation of Gram-negative bacteria in the gastrointestinal area leads to the production of harmful metabolites (i.e., endo- and exotoxins, saccharides, and amyloids) [46]. Exotoxins, such as LPS, which have the ability to activate the production of pro-inflammatory cytokines, exert negative effects. LPS belongs to glycolipid molecules, mostly produced by Gram-negative bacterial strains. These molecules participate in the integrity maintenance of the bacterial outer-membrane permeability barrier, playing an important role in host-pathogen interplay [47]. The overproduction of bacterial LPS can cause the activation of the macrophage–monocyte nod-like receptor P3 inflammasome, therefore stimulating the production of cytokines with pro-inflammatory activity, such as IL-18 or IL-1*β*. LPS can be translocated and intensify neuroinflammatory diseases such as AD [48]. Additionally, it was demonstrated that IL-1*β* can decrease the phagocytic function of microglial cells, stimulating the hyperphosphorylation of tau protein, and causing reduced synaptic plasticity and cognitive deterioration [48,49,50]. These cytokines can induce microbiota dysregulation, increasing gut permeability and influencing gut bacteria diversity [51,52].

On the other hand, the gut microbiome is able to produce a broad spectrum of metabolites, which have beneficial effects on human health (Figure 2).

Recently, bacterial strains producing SCFAs have gained extreme popularity. SCFAs are small organic monocarboxylic acids, released after the fermentation of indigestible alimentary fiber (mainly galacto- or fructooligosaccharides), plant-derived polysaccharides, or the metabolism of amino acids, and are considered a key gut-derived metabolites with beneficial health effects and as essential in gut-brain crosstalk [53,54]. The type of formed SCFAs is correlated with the kind of fiber ingested and the overall gut microbiota population. For example, microbes belonging to *Firmicutes* phyla (e.g., *Lachnospiraceae*, *Eubacterium*, or *Roseburia* belonging to the *Clostridia* class) are responsible for the main production of butyrate, whereas *Bifidobacteria* spp. produce lactate and acetate [55]. SCFAs are significantly reduced during dysbiosis, where disrupted gut epithelial integrity, inflammation, and altered microbiome metabolic processes occur [56,57,58]. In a healthy colon, levels of SCFAs can vary depending on a diet; however, in multiple diseases, these levels are altered. SCFAs may also act in an indirect or direct way through G-protein-coupled receptors or as histone deacetylase epigenetic modulators [59]. They participate in various physiological processes, such as cell energetics and colonocyte metabolism and skeletal, adipose, and liver tissue modulation [60]. Their major functions comprise the activation of trophic factors, energy management, and the manufacturing of regulatory T-cells [61]. Short-chain fatty acids can be considered as key intermediaries in numerous neurological diseases such as Parkinson’s and Alzheimer’s diseases, stroke, and neuropsychiatric conditions.

### 3.1. Alzheimer’s Disease

Alzheimer’s disease (AD), commonly known as senile dementia, is a chronic, devastating, neurodegenerative, and neuroinflammatory disorder with rising incidence and is considered to be the most frequent form of dementia in aging individuals (caused by aberrant protein processing, trafficking, and aggregation in neural cells). The World Alzheimer Report indicates that 50 million people suffer from AD-related neurological problems and this number will increase to 152 million in 2050 [48]. It is predicted that in 2030 up to 66 million people will suffer from AD [9,35,62]. AD is a multifactorial disease characterized by synapse loss, extracellular cerebral accumulation of insoluble peptides belonging to amyloid-*β* (A*β*), the aggregation of hyperphosphorylated tau protein inside the cells, neurofibrillary mass formation, neuronal death (in the neocortex and hippocampus), progressive memory deterioration, and cognition decline leading to dementia. AD is associated with more than 700 genetic and environmental risk factors. From the genetic point of view, one of the most important risk factors related to AD development is the incidence of a “4 allele variant in the apolipoprotein” (*APOE4*) gene, which is associated with cholesterol transport in CNS, and also contributes to metabolomic and taxonomic changes in the microbiome [63,64].

#### 3.1.1. Host and Microbiome Metabolomic Changes during AD: Short Story about Bad and Good Cop

##### Good Cop

As previously mentioned, the microbiome is able to produce SCFAs. Its indirect influence on Alzheimer’s disease progression has been reported in multiple studies [65]. The gut microbiota of AD patients represented a diminished amount and incidence of bacteria producing butyrate and proinflammatory-acting bacterial taxa [66]. Many studies revealed that the decreased levels of SCFAs can cause increased gut permeability together with altered gut pH levels, enabling the spread of opportunistic bacteria from *Shigella* and *Escherichia* species, elevated both in AD and PD [67]. Studies on the Japanese population underlined the negative correlation between SCFAs and the onset of AD and type 2 diabetes [68].

The main end product of bacterial fermentation, and therefore the most predominant, is the SCFA propionic acid (PA), which positively influences CNS regeneration processes such as remyelination (due to the former increase of the amount of intestinal T regulatory cells and protecting against the white matter and axonal loss) [69]. Hoyles et al. demonstrated the contribution of propionate to the BBB integrity and the modulation of gut-brain axis [70]. BBB homeostasis and integrity are necessary for proper CNS development and function. Recently, a positive correlation between microbial-induced BBB dysfunction (barrier integrity disturbances followed by leakage) and disorders such as AD has been established [3,71]. Another member of the SCFAs family, which has a documented influence on homeostasis and AD progression, is butyrate, an important driver of metabolic processes with the ability to influence intestinal macrophages, increasing immunomodulation and decreasing the histone deacetylase inhibition-mediated production of pro-inflammatory cytokines [72]. Butyrate, as an SCFA, influences the immune response crossing the BBB and drives the maturation and metabolism of microglia cells. It regulates the secretion of gut hormones, such as glucagon-like peptide-1, which can improve hippocampus neuroplasticity [38,56,73]. Additionally, in vitro tests conducted by Ho et al. revealed that bacterial SCFAs, such as valeric and butyric acid, exhibited a strong inhibitory effect on amyloid-*β* aggregation, which can be absorbed in the intestines and have a negative influence on disease progression [74].

Low concentration of SCFAs means a reduced degree of acetylation, therefore causing chromatin remodeling changes and low levels in plasma. An inadequate, low-fibre diet can seriously disturb the intestinal flora, as well as cause disproportions in the quantity and quality of its metabolites. The abovementioned metabolites include SCFAs, secondary bile acids (products of cholesterol metabolism and clearance), lipids, and vitamins [75,76].

Besides SCFAs, the human microbiome is able to produce other small molecules with bioactive potential. The species of the genera *Bifidobacterium* and *Lactobacillus* produce various metabolism products such as GABA, dopamine, histamine, serotonin, tryptophan catabolites, and noradrenaline, among others, with all of them belonging to neurotransmitters that regulate cognitive functions. Experimental studies have confirmed that bacterial strains such as *Bifidobacterium* and *Lactobacillus* can produce the GABA neurotransmitter through *L*-glutamate decarboxylation [77,78]. Neurotransmitters such as serotonin and dopamine can affect the immunological response, whereas GABA protects against bacterial translocation and is considered to be a major inhibitory neurotransmitter in the human CNS [79,80]. Moreover, they fulfill the role of modulators of neurodegenerative disease severity, mainly through immune-mediated neurodegeneration or microbial metabolites directly influencing neural cells. It was demonstrated that the aforementioned metabolites are the main signaling molecules for microbial effects on gut–brain communication and their disturbances are linked with gut dysbiosis and cognitive impairment [39]. For example, the results of metabolic profiling in AD patients indicated the disturbances in serotonin metabolic pathway metabolites, which were significantly decreased, especially 5-hydroxytryptophan and DL-5-methoxytryptophan, are mainly produced in the gut by enterochromaffin cells [48]. This process probably can be related to the disruptions in intestinal microbiota content, in particular to the reduction of *Clostridium sporogenes* and *Ruminococcus gnavus* abundance, which are considered the main tryptamine producers. Tryptamine impacts gut conditions through its action as a *β*-arylamine neurotransmitter capable of inducing serotonin (5-hydroxytryptamine) release by intestinal mucosal surface cells [81,82]. Several bacterial species, such as *Escherichia coli*, *Clostridium* spp., *Bacteroides* spp., *Lactobacillus* spp. *Peptostreptococcus* spp., or *Vibrio cholerae* are characterized by the ability to convert tryptophan into indole and indole derivatives through the tryptophanase enzyme, produced both by Gram-negative and Gram-positive bacteria [83,84]. Among them, 3-methylindole (skatole), indoleacetic acid, indoleacrylic acid, indolealdehyde, indolelactic acid, indolepropionic acid, or tryptamine can be found [85]. Interestingly, these metabolites are the ligands of the aryl hydrocarbon receptor (AHR), which can boost the innate and adaptive immune responses, being an important transcription factor expressed by immune cells [86]. Moreover, they can stimulate gut hormone secretion, and gastrointestinal motility, and improve the epithelial barrier of the intestine, regulating intestinal redox homeostasis [87]. Additionally, microbial tryptophan catabolites possess anti-oxidative and anti-inflammatory properties [85]. Previous scientific reports suggested that microbiota diversity impacts the tryptophan bioavailability and its downstream metabolites, which can participate in the host-microbiota crosstalk, being important intracellular signaling molecules [85]. This observation could be a result of disrupted homeostasis in the gut. In recent years, researchers have established a link between disturbed tryptophan metabolism (including indole derivatives and serotonin synthesis) and intestinal dysbiosis. Additionally, Huang et al. demonstrated that indole-3-propionic acid (tryptophan gut bacterial metabolite) could be used as a promising marker of AD, indicating progressive cognitive impairment correlated with disrupted gut microbiota composition and reduced indole metabolite concentration [88,89,90].

##### Bad Cop

When we talk about negative metabolites or microbiota by-products, which have a documented impact on AD progression, two molecules have been in the spotlight during the past decade. One is LPS and the second is gut-derived amyloid-*β*, both acting as pro-inflammatory agents.

Some of the studies performed show a connection between the pathological microbiota and Alzheimer’s disease. Screening tests identified the connection between *Escherichia* and *Shigella* bacterial species and increased systemic pro-inflammatory cytokine-related processes, reported in patients with Alzheimer’s. Increased inflammatory processes and oxidative stress conditions can trigger neurodegeneration observed in people with Alzheimer’s disease [11]. The previously mentioned species belong to Gram-negative bacteria, which are capable of producing huge amounts of pro-inflammatory lipopolysaccharides (LPSs), thereby causing intestinal inflammation and gut barrier disruption. Increased gut permeability permits the transport and the systemic and brain accumulation of LPS or other bacterial toxins and the propagation of an inflammatory state [91]. Moreover, this observation has been confirmed in *in vivo* studies, where the injection of bacterial lipopolysaccharide into the brain imitates inflammatory and pathological features such as cognitive impairment and fibrillogenesis observed in AD [92].

Besides the propagation of an inflammatory state, microbial endotoxins are probably involved in AD’s inflammatory amyloidosis. Metagenomics analysis showed that patients suffering from cognitive dysfunctions related to amyloidosis demonstrated elevated amounts of pro-inflammatory cytokines and neurotoxic compounds. In these patients, increased inflammation was probably related to *Eubacterium* deficiency, and an overabundance of gram-negative *Escherichia* and *Shigella* species [93], which are known amyloid producers [74]. They manufacture “curli” amyloid fibers, accumulated extracellularly through an elaborated system [94]. Further studies revealed that the overproduction of A*β* plaques is correlated with excessive growth of *Escherichia coli*, *Bacillus subtilis*, *Klebsiella pneumoniae*, *Mycobacterium* and *Salmonella*, *Staphylococcus aureus*, and *Streptococcus* spp. [95]. Galloway et al. found A*β* in small intestine epithelial cells, suggesting that gut-derived amyloids could be absorbed and finally reach systemic circulation, where they can act as pro-inflammatory factors and trigger fibrillogenesis [96,97]. Additionally, it was shown that amyloids generated by microbes can influence IL-17A and IL-22 proinflammatory cytokines production, considered as a key players in NF*κ*B signaling, both associated with AD and capable of crossing the BBB and GI tract [9].

Another Gram-negative bacteria, and its metabolites, which have a confirmed association with AD, is *Helicobacter pylori*. *H. pylori* mainly colonizes the stomach, where it is able to produce H_2_O_2_, leading to increased homocysteine levels, and is a well-known metabolite connected with AD development. H_2_O_2_ alone harms BBB vascular endothelial cells, thereby disrupting cell homeostasis and creating dysfunctions, and consequently increasing the *β*-amyloid concentration in the brain [19,48,98].

At the end of this chapter, it is worth mentioning the fact that the interaction between the microbiome and the host is bidirectional, and some metabolic changes, which could be a pathological result of disease progression, also influence microbiome metabolism. A good example is a disturbance in cholesterol homeostasis, which correlates with an augmented probability of Alzheimer’s disease occurrence [63,64]. In addition, alteration of the intestinal microbiota composition with the subsequent fluctuations in the serum and brain levels of bile acids (BAs) is involved in the development of AD. BAs can act as detergents, affecting BBB permeability, and incrementing brain influx of peripheral cholesterol and BAs. Therefore, proper cholesterol elimination pathways play a crucial role in AD prevention and treatment [50]. The gut microbiome is a main “organ” metabolizing secondary bile acids. Peripheral cholesterol is eliminated mainly through its liver conversion to BAs. The gut microbiota performs BA transformation via deconjugation and dehydroxylation, leading to secondary BA formation [99]. Nho et al. investigated the correlation between neurodegeneration and BAs, *B*-amyloid, and tau protein concentration. The application of targeted metabolomics enabled the determination of the association between intestinal microbiota-derived secondary bile acids and sera BAs. Interestingly, the analysis indicated an imbalanced BA ratio, underlying an increased amount of secondary BAs over primary ones [91]. Another clinical study confirmed the correlation between bile acid metabolic profile and the prevalence of AD and mild cognitive impairment, showing similar BA ratio disruptions. Decreased cognitive functions were positively associated with increased concentrations of secondary bile acids such as deoxycholic acid, glycodeoxycholic acid, taurodeoxycholic acid, and glycolithocholic acid. The described results suggested the involvement of the pathological gut microbiota in AD development; however, still it requires more in-depth studies [100].

### 3.2. Parkinson’s Disease

Parkinson’s disease (PD) is one of the most widespread neurodegenerative diseases, second place in incidence among them, concerning around 1% of the population over the age of 60, reducing the quality of life and leading to progressive disability [101]. The death of dopaminergic neurons in the *substantia nigra pars compacta* is characteristic of PD. This disease causes symptoms such as rest tremors, bradykinesia, muscular rigidity, and postural stability problems. Parkinson’s disease is characterized not only by neurological disorders, but also by gastrointestinal disturbances, including nausea, constipation (affecting more than 70% of PD patients), vomiting, reduced colonic motility, and gut microbiota changes (small intestinal bacterial overgrowth among many others). Additionally, gastrointestinal comorbidities usually appear years prior to the onset of motor function symptoms [102,103].

The origins of Parkinson’s disease are the result of many factors such as increased age, polygenic factors (disease-segregating mutations in *α*-synuclein protein occurring in 15% of PD patients), toxins, and infectious agents. In 2006, Braak et al. assumed the gut-related origins of Parkinson’s disease [104]. One of the key pathological hallmarks of PD is Lewy bodies, being eosinophilic inclusion forms of misfolded *α*-synuclein aggregates. Importantly, *α*-synuclein protein has a tendency to misfold and form aggregates. Its accumulation has been detected during bacterial and viral infections in the gut. Moreover, it can be recognized as a characteristic indicator of PD neural degeneration [105]. This protein aggregate can boost the local immune response [106]. Interestingly, *α*-synuclein deposition has been confirmed in the neurons of the intestinal submucosa, together with a possibility of the *α*-synuclein transport to the brain via the vagus nerve (translocation of prion-like proteins) [103,104,107,108]. Misfolded *α*-synuclein triggers neuronal dysfunction and progressive degeneration, thereby enabling disease progression [109]. Additionally, *α*-synuclein deposition in CNS is associated with intestinal hyperpermeability, TLRs, and pro-inflammatory cytokine overexpression. Animal models showed that *α*-synuclein aggregation in PD is associated with disorders of lipid metabolism, especially concerning phospholipids and sphingolipids [110,111].

#### 3.2.1. Host and Microbiome Metabolomic Changes during PD: Short Story about Bad and Good Cop

##### Good Cop

Patients suffering from PD have different bacterial flora compared with healthy controls [112]. Microbiota have an impact on disease progression, which has been established in the fecal-transplantation experiments. Administration of the microbiota of patients with Parkinson’s disease to mice showed the development of neuroinflammatory processes and motor deficits [113].

The PD microbiome has a diminished variety of bacterial taxonomic groups, especially the ones associated with anti-inflammatory and neuroprotective effects and the increase of phyla, which has toxic effects [114,115]. Predominantly, the reduction was observed in the *Lachnospiraceae* family, including *Butyrivibrio*, *Pseudobutyrivibrio*, *Coprococcus, Blautia* and SCFAs-producing *Prevotellaceae* [115]. The increased amount of *Enterobacteriaceae* is linked to the severity of postural instability and walking difficulties [116]. Moreover, the PD microbiome is characterised by increased *Akkermansia* and *Bifidobacterium* species and decreased amount of *Faecalibacterium* and *Lachnospiraceae*. The overgrowth of *Akkermansia* spp. in PD patients may be strain-specific and can be related to immune-response alterations, decreased mucus thicknesses, and increased constipation [117]. *Faecalibacterium* family is a crucial player producing SCFAs and anti-inflammatory metabolites, maintaining gut health [118]. *Faecalibacterium* balance disturbances can lead to impaired gut-barrier function, increasing the risk of infection with enteric pathogens and boosting *α*-synuclein formation [119]. Similarly, the *Lachnospiraceae* family can produce butyrate, important for the proper function of the gut epithelium [120]. Decreased *Lachnospiraceae* quantity might aggravate gut inflammation, stimulate toxin production, and damage the integrity of the gut epithelial barrier [121,122]. PD patients’ intestinal microflora were characterised by a diminished abundance of *Lachnospiraceae*, which was positively correlated with PD progress, cognitive decline, and impaired motor functions [122,123]. These microbiome’s compositional changes correlate well with some metabolic switches. PD microbiota show reduced SCFAs production, decreased carbohydrate fermentation, and augmented proteolytic fermentation [102]. Chen et al. demonstrated that the SCFA level was relevant to microbiota composition changes and the clinical severity of PD. The levels of SCFAs acetate, butyrate, and propionate in fecal samples of PD patients were significantly reduced, contrary to their elevated concentration in plasma [57]. Rodent models of PD disease have shown a positive effect of SCFAs on behavior and PD symptoms [11,124]. Butyrate, as a histone deacetylase inhibitor, exerts neuroprotective activity, reducing dopaminergic cell death, and motor impairment, and improving cognitive functions [125,126]. Taken together, SCFAs play an important role in protection from PD disease and could be proposed as representative gut-oriented indicators for diversifying individuals with PD and placing disease progression [127].

During PD development, microbiota changes can be accompanied by reduced concentrations of branched-chain amino acids (BCAAs) and aromatic amino-acids in comparison with the healthy control group [128]. It has been demonstrated that the alteration of the metabolites occurred, especially during disease progression [129]. Experimental results indicate the correlation between PD and mitochondrial dysfunction. Additionally, the lower levels of BCAAs and aromatic amino acids have been observed in stool samples during this period in comparison with the healthy control group, indicating a dysbiosis state in the gut microbiota [128]. Moreover, the PD-disrupted gut microbiome may negatively influence the level of methionine in human plasma [130,131].

Similar to the AD case, the host metabolome is affected during disease, which is not without its influence on microbiome composition/metabolome. Bile acid synthesis is a common part of the metabolic system which is affected both in AD and PD [132,133,134]. Regarding this, there were increased levels of bile acids conjugated with taurine (taurodeoxycholic, taurolithocholic, and taurochenodeoxycholic acid) observed, which correlates with motor activity in PD [135]. It is worth underlining the role of taurine as an inhibitory neurotransmitter. Moreover, it is fabricated and secreted by neurons during stress conditions, e.g., mitochondrial dysfunctions, where taurine seems to increase the neuronal survival rate due to calcium influx regulation and the intensified antioxidant gene expression. In summary, taurine demonstrates an important role in proper brain function and development. Additionally, it contributes to the volume regulation of brain cells, and interferes with synaptic amino acid receptors [136,137].

##### Bad Cop

Functional analysis of the samples obtained from PD patients exposes an augmented microbial ability to conduct the degradation of mucin and glycans [138], influence on folate deficiency and hyperhomocysteinemia [138], and finally increased proteolytic fermentation with the creation of toxic amino acid metabolites such as *p*-cresol or phenylacetylglutamine [102]. Preclinical tests on genetically modified mice with PD demonstrated the correlation between *Akkermansia muciniphila* and *Bilophila wadsworthia* abundance and decreased the mice motor activity. Furthermore, *Akkermansia muciniphila* produces high amounts of pro-inflammatory hydrogen sulphide, correlated with PD pathology and being a neurotoxin affecting brain mitochondrial energy homeostasis through the inhibition of brain glutathione concentration [135,139]. The second bacterial species, *Bilophila wadsworthia*, is considered to be the main producer of sulphite in the human gut microbiome, leading to intestinal inflammatory states and the reduction of blood glutathione levels [135].

### 3.3. Other Neurodegenerative and Psychiatric Diseases

The term “neurodegenerative diseases” is much wider than only PD and AD [140]. However, according to the state of knowledge, microbiome, and microbiome–metabolome influence on disease progression in these two diseases are the best described so far (Table 1). Interestingly, microbiome and microbiome–metabolome have gained a lot of attention recently [141,142], and new studies shed light on their engagement in multiple sclerosis (MS), amyotrophic lateral sclerosis (ALS), and Huntington’s disease (HD) [143].

### 3.4. Microbiota, Multiple Sclerosis (MS), and Amyotrophic Lateral Sclerosis (ALS)

Multiple sclerosis is a chronic neurodegenerative autoimmune disorder, where the human immune system attacks the myelin sheath surrounding axon terminals, thereby triggering long-lasting inflammation within the CNS, abrasion formation, and demyelination of axons. Besides, afterward, it can cause autonomic and cognitive problems, motor, sensory, and visual flaws, and finally paralysis and disruption of the BBB [144,145,146]. Various animal and human studies suggested the participation of gut microbiota in the development of MS [147,148,149]. Animal studies on an experimental autoimmune encephalomyelitis mice model revealed a significant reduction in *Lactobacillus* phyla, contributing to an impairment of the mice’s immune systems [150]. In patients diagnosed with MS, a reduction in the variety and quantity of gut bacteria was detected, confirming a moderate dysbiosis state [151]. Fecal samples obtained from MS patients indicated the reduced amount of bacteria belonging to 19 species, *Bacteroidetes*, *Firmicutes, Faecalibacterium*, *Prevotella*, and *Anaerostipes* species among them, and the increased quantity of *Actinobacteria, Bifidobacterium*, and *Streptococcus* genera [146].

Experimental research confirms the hypothesis regarding SCFA reduction in autoimmune diseases (MS among others) and altered gut microbiota. A low level of SCFAs dysregulates T_reg_ cell functions, leading to insufficient immune tolerance towards endogenous components and enteric microbiota. However, this process can be reversed by SCFAs supplementation (e.g., propionic acid) [147,152]. Furthermore, research results indicated that MS is also connected with intestinal permeability disruption and the altered metabolism of bile acids, causing the dysregulation of the immune and nervous systems. Bile acids can act as CNS inflammatory signalling modulators, e.g., ursodeoxycholic acid can exert microglia inflammatory-inhibiting activity. Moreover, tauroursodeoxycholic acid (TUDCA) can direct microglia phenotypes towards an anti-inflammatory state, reducing NF*κ*B activation and exerting a neuroprotective effect both in the mouse model of acute neuroinflammation and on microglial cells in vitro. TUDCA binds to the G protein-coupled bile acid receptor 1/Takeda G protein-coupled receptor 5 (GPBAR1/TGR5) expressed in microglial cells [153]. This binding follows an increased level of microglial intracellular cyclic adenosine monophosphate (cAMP), inducing the release of the anti-inflammatory cytokines and chemokines (e.g., IL-4Rα, IL-10, IL-1 receptor-associated kinase-M IRAK-M, programmed cell death ligand 1 PD-L1, and sphingosine kinase 1 Sphk1) and reducing the ones with pro-inflammatory activity (ionised calcium-binding adapter molecule 1 Iba-1, inducible nitric oxide synthase *i*NOS). Experimental results demonstrated that TUDCA acted as a double-edged sword, triggering microglia enhancement in anti-inflammatory and alternatively activated markers and on the other hand reducing the infiltration of macrophages and microglia with expressed pro-inflammatory markers [153]. Interestingly, during neuroinflammation, microglial cells change their morphology, converting themselves into reactive macrophage-like forms. However, microglial cells are flexible, as they can sometimes polarize towards alternatively activated phenotypes, promoting CNS pathologies (proinflammatory or M1 microglia), or exert anti-inflammatory effects, boosting cell repair and remodeling. Bile acids were identified as nuclear hormone receptor farnesoid X receptor agonists, leading to experimental autoimmune encephalomyelitis (EAE) depletion (in a mouse model) and neuroinflammatory process adjustment [153,154]. The administration of the complex probiotic positively influenced the intestinal flora balance in patients diagnosed with MS. The probiotic prevented the onset of dysbiosis, significantly increasing the amount of *Methanobrevibacter* and *Akkermansia*, *Prevotella*, and *Sutterella* in MS patients, influencing gene expression and the production of beneficial bacterial metabolites [155].

Amyotrophic Lateral Sclerosis (ALS) is a fatal health condition caused by the progressive neural death of the spinal cord, brain stem, and motor cortex neurons, causing advanced weakness, paralysis, eating difficulties, and respiratory failure [156]. One of the main signs of ALS is the accumulation of phosphorylated protein, which binds to DNA in the form of protein aggregates deposited in glia and motor neurons [157,158]. Next-Generation Sequencing (NGS) methods, such as 16S rRNA gene sequencing, have demonstrated the alteration of numerous bacteria, including *Parabacteroides distasonis*, *Lactobacillus gasseri*, *Prevotella melaninogenica*, *Ruminococcus torques*, and *Akkermansia muciniphila* [159]. Changes in bacterial abundance and diversity correlate with ASL origins (as one of the environmental factors) together with impaired metabolism, immunological functions, and toxin accumulation, leading to brain damage. It has been demonstrated that *Akkermansia muciniphila* alleviates ALS symptoms, although *Ruminococcus torques* and *Parabacteroides distasonis* aggravate them [157]. The main switches have been observed in SCFA-producing bacteria, leading to reduced levels of SCFAs, e.g., butyrate and propionate [157,158].

### 3.5. Huntington’s Disease (HD)

Recent studies underlined the important role of the gut microbiota in bidirectional crosstalk in Huntington’s disease (HD) [160]. Huntington’s disease has genetic origins related to trinucleotide expansion (CAG) in the huntingtin coding gene. Moreover, experiments performed in mice models revealed that the altered microbiota observed in the HD-suffering individuals at the pre-motor symptomatic stage highly influence health status. HD microbiota composition indicated dysbiosis, disturbances in cytokine levels, and an increase in sulphur metabolism characterized by hydrogen sulphide overproduction and negatively influencing gut health [160,161]. Ultimately, the gut dysbiosis state was also described in a transgenic mice model representing Huntington’s disease [162,163]. Kong et al. found increased *Bacteroidetes* and decreased *Firmicutes* levels in a R6/1 transgenic mice model. In this study, dysbiosis was followed by body weight deterioration, gastrointestinal motility problems, and the loss of motor ability [162]. Subsequent studies performed by the same group revealed the negative correlation between *Blautia producta* and butyrate production. Moreover, *Prevotella scopos* was negatively associated with ATP levels [161]. Clinical trials performed on patients suffering from Huntington’s disease revealed the prevalence of bacterial phyla belonging to *Firmicutes* (83%), *Actinobacteria* (9%), *Bacteroidetes* (4%), and *Verrucomicrobia* (1.1%) detected in the collected fecal samples [164]. The results of performed human randomized clinical trials positively correlate with the data obtained in mouse models in which gut microbiota dysbiosis was observed. A Huntington’s mouse model revealed endocrine hormonal abnormalities, improper intestinal morphology (decreased mucosal thickness, and villus length), and damaged neuropeptide production [165,166,167].

### 3.6. Autism, Schizophrenia, and ADHD

Autism spectrum disorder (ASD) can be classified as a neurodevelopmental problem represented by repetitive behaviors, social communication, and cognitive function impairment beginning in early childhood [168]. It is predicted that more than 50% of the neurobiology in autism disorders may be determined by non-genetic factors, such as environmental factors, parental age, and preterm birth, among many others. Autism is not only associated with disorders of a neurological nature, but also with disturbances in the qualitative and quantitative arrangement of the gut microbiota, intestinal immune inflammation, and dysbiosis, as confirmed in many studies, and correlated with disease severity [169,170]. For example, gut microbiota transplantation from patients suffering from autistic disorders into antibiotic-treated mice induced pathological autistic social and repetitive behaviors in rodents [11,171].

In accordance with the microbiome–metabolome, research has shown that reductions in microbiome diversity were mainly expressed by the decrease of the SCFAs-producing *Prevotella*, *Coprococcus*, *Faecalibacterium*, and unclassified *Veillonellaceae* species [172,173]. However, it has been found that ameliorating and modulating these behaviors by administering probiotics containing *Bacteroides fragilis* or *Lactobacillus reuteri*, or by performing fecal transplantations from healthy individuals, is possible. In this way, the composition of the beneficial gut microbiota can be restored, providing adequate levels of SCFAs and intestinal homeostasis [11,171].

Schizophrenia is a severe neuropsychological disorder affecting approximately 1% of the world’s population [174]. 16S rRNA sequencing and diversity analyses of the fecal-derived microbiota obtained from patients suffering from schizophrenia indicated lower microbiome diversity. Performed analyses revealed the abundance of *Proteobacteria* and decreased amount of SCFAs synthesizing bacteria from *Blautia*, *Coprococcus*, and *Roseburia* species. It has been reported that butyrate and other SCFAs can cross the gastrointestinal endothelium and can pass through the BBB (exploiting active membrane transporters) and inhibit histone deacetylase 1 (HDAC1) [6]. Abnormal microbiota can lead to reduced levels of butyrate and can cause the elevation of HDAC1 levels in the prefrontal cortex and hippocampus of patients with diagnosed schizophrenia, causing altered DNA methylation, impaired synaptic plasticity and short-term memory, and the development of impaired cognitive function and social abilities [175].

The analysis of the intestinal microbiome of adolescent and adult ADHD patients (Attention Deficit Hyperactivity Disorder) indicated changes and imbalances in the gut microbiota followed by elevated levels of *Bifidobacterium* and *Eggerthella* genera related to boosted dopamine precursor formation and connected with ADHD development [176]. Some reports indicated the implication of the microbiome in the regulation of brain dopamine levels [177]. The *Bifidobacterium* genus plays an important role in dopamine precursor production (cyclohexadienyl dehydratase) in ADHD patients [178]. Jiang et al. reported a reduction of *Faecalibacterium* spp. in paediatric patients diagnosed with ADHD and they suggested *Faecalibacterium* as a new marker of ADHD [179]. *Faecalibacterium* depletion was followed by increased cytokine levels and inflammation [180,181]. Interestingly, three bacterial species *(Bacteroides uniformis*, *Bacteroides ovatus*, and *Sutterella stercoricanis*) were identified as potential biomarkers of ADHD [182]. Tengeler et al. reported increased anxiety in a GF mice model with human microbiota transplanted from ADHD-suffering individuals [183]. Another clinical trial performed on infants indicated the beneficial and protective effect of *Lactobacillus rhamnosus* GG administration in the prevention of ADHD development. Moreover, *L. rhamnosus* probiotic administration reduced the quantity of the following proinflammatory factors: IL-12 p70, IL-10, TNF-*α*, and IL-6 [184].

### 3.7. Brain Injury and Stroke

It has been proven that risk factors for civilization diseases such as stroke and cardiovascular diseases are also associated with the diversity of the intestinal flora. The abovementioned factors comprise atherosclerosis and hypertension. Moreover, the dysregulation of microbiota was observed in patients with stroke together with increased plasma and urinary concentrations of the proatherosclerotic gut bacterial metabolite trimethylamine *N*-oxide (TMAO), also co-responsible for the development of cardiovascular diseases (atherosclerotic coronary artery disease among others) [185], AD, and gestational diabetes [186]. TMAO is produced from dietary phosphatidylcholine by the intestinal microbiota [11]. Preclinical models of cerebral ischemia indicate the coexistence of pathological microflora which negatively affect intestinal permeability and motility. Some reported experimental results indicate the relationship of intestinal microbiota with neuroinflammatory processes by modulating intestinal T-cell trafficking to the CNS. Animal models of cerebral ischemia-reperfusion injury clearly pointed out the beneficial, neuroprotective effect of the administration of a probiotic containing the *Clostridium butyricum* bacterial strain on the prognosis of animals after a stroke [187].

## 4. Nutritional Intervention as a Promising Solution to Prevent the Progression of Neurodegenerative Diseases

Currently, we have defined multiple factors which have an influence on the microbiome’s composition and its metabolome, i.e., environmental (such as exposure to pesticides or heavy metals), biological (infections), and sociodemographic factors such as inadequate diet, stress, mode of delivery at birth, lack of breastfeeding during the neonatal period, or antibiotic therapy [161,188,189]. All of these factors could negatively modulate the microbiome and finally lead to dysbiosis. However, if we try to indicate the most powerful factor, then diet is one of the strongest lifetime modulators of microbiome composition [190,191].

For example, the Western diet is a well-known negative modifier of microbiome composition and a promoting factor of multiple common diseases [192]. It is characterized by high fat and sugar ingestion and low fiber content, which are positively linked to *Bacteroidetes* and *Actinobacteria* abundance, but negatively correlated with fiber-rich nourishment. A contradictory connotation could be observed for *Proteobacteria* and *Firmicutes* [193].

Contrarily, as a positive example, balanced diets (i.e., Mediterranean diet and Dietary Approaches to Stop Hypertension) can be mentioned. Diets abundant in natural compounds with anti-inflammatory activity, antioxidants, polyunsaturated fatty acids, and plant-origin nutrients (proteins, polyphenols, vitamins, and fibers), together with reduced caloric intake, such as the Mediterranean diet, are correlated with a lower risk of dementia, decrease age-related cognitive deterioration, and the risk of neurodegeneration occurrence [194,195,196]. Moreover, the Mediterranean diet increases the amount of gut *Bifidobacterium* and *Lactobacillus* and reduces malignant bacteria belonging to *Clostridium* and *Bacteroides*, thereby improving memory and cognitive processes in healthy females after menopause and disorder patients. Additionally, following such a diet significantly reduces amyloid aggregation and therefore the frequency of amyloid-related diseases in the aging population [14,197,198]. There are scientific reports that he gut dysfunction is related to the inflammation of the CNS, which can especially be observed in Parkinson’s disease patients, where dysregulation of the gut–brain axis was observed in 80% of patients [199].

In the past decade, single dietary factors which can positively manipulate the gut microbiota–brain axis through microbiome modulation have gained significant attention [200,201]. One of these factors is probiotic bacteria and/or prebiotics. Probiotics are defined as microbial organisms with a beneficial role in the host’s health and homeostasis, which help to preserve digestive, metabolic, immune, and neuroendocrine functions [202]. Naturally probiotic bacteria can be found in fermented food [203,204]. Experimental results revealed that the administration of probiotics alone or in combination with prebiotics (fructooligosaccharides or galactooligosaccharides) improved the gastrointestinal barrier [205,206], which is the first line of “defence” from harmful factors occurring in food, and the microbiome. Probiotics and prebiotics are well-documented “problem-solution” for gastrointestinal diseases [207,208]. Moreover, probiotics can reduce CNS inflammatory processes and activation of microglia, demonstrating promising potential for use in neurodegenerative diseases as ingestible “psychobiotics”. Many studies indicate that the ingestion of probiotics can serve as a microbiological strategy against neurodegenerative diseases such as AD [209,210,211]; however, the exact mechanism is under investigation [212]. Experiments performed on mice underline the hypothesis that mechanisms related to neuroprotective effects of *Bifidobacterium breve* administration may be connected with neuroinflammation and cognition pathways, probably through the production of the brain-derived neurotrophic factor (BDNF) neurotransmitter and the regulation of the gut microbial composition [213]. Clinical results of a randomized double-blind controlled trial revealed that 3-month exposure to *Lactobacillus* spp. And a *Bifidobacterium bifidum* “cocktail” enhanced cognitive functions and memory, as examined in a mini-mental state test [214]. It was demonstrated that *Bifidobacterium breve* exerts positive and unique activity in AD patients, improving cognitive functions and reducing neuroinflammation. Controlled, double-blind, randomized clinical trials with a 12-week multispecies probiotic treatment (comprised of *Lactobacillus acidophilus*, *Lactobacillus casei*, *Bifidobacterium bifidum*, and *Lactobacillus fermentum*) caused a significant decrease in malondialdehyde in the probiotic-treated group [214].

### 4.1. Clinical Trials Related to Probiotic Administration and Brain-Related/Stress Disorders

Actually, some clinical trials regarding probiotic administration and brain-related disorders have been carried out. Herein, we gathered information regarding this issue. An intense clinical trial search was performed between 16 September and 19 September 2022 on the National Library of Medicine (NIH) ClinicalTrials website (clinicaltrials.gov) using the term “probiotics” in the “condition” field, yielding 853 clinical studies. Only completed studies were taken into consideration. After the initial selection, we collected only those trials related to brain related/stress disorders (Table 2). Interestingly, the majority of the presented clinical trials demonstrated a positive correlation between probiotic administration and the amelioration of disease/condition symptoms, encouraging the daily administration of pro/symbiotics or the introduction of fermented-food additives into the daily diet.

### 4.2. Another Factors including Gender Aspects Related to Gut Microbiome and Neurodegenerative Diseases

The health of the gut microbiome can vary depending on various factors including diet (e.g., ingestion of saturated/unsaturated dietary fats, vitamins, minerals, dairy products, alcoholic beverages, polyphenols containing food such as cocoa-rich chocolate, tea, or coffee), dietary patterns, geographical area, and diet-related habits (sometimes varying between males and females) and sex (the influence of sex hormones). A positive correlation between polyunsaturated fatty acids (PUFA), omega 3 polyunsaturated fatty acids (N-3 PUFA), and *α*-linolenic acid (aLNA) ingestion and protection against PD development in dose-response trends was demonstrated [253]. American studies demonstrated that PUFA replacement with increased animal fat intake (Western Diet) significantly increased the risk of PD in males only [254]. Excessive animal fat intake can promote dysbiosis and the production of harmful microbial metabolites, which can increase the prevalence of the development of neurodegenerative diseases. Moreover, sex differences are implicated in the mental health and prevalence of psychiatric, neurodevelopmental, and neurodegenerative disorders [255]. It has been demonstrated that PD patients usually have altered microbiome compositions compared with controls [120]. Similar to AD, many sex differences in PD are correlated with the protective nature of estrogens, as they are impacted by the gut-brain axis and protect against oxidative damage, thereby supporting dopaminergic function [256]. It has been demonstrated that at age 45, the estimated overall lifetime risk for AD is about 20% for females and 10% for males, which is followed by a disruption in the patient’s gut microbiota composition [257]. In summary, it is worth underlining the correlation between the gut microbiome, immune system and brain, mental health, and behavior [258].

## 5. Conclusions

Overall, gut microbiota and their implication in the gut-brain axis can be recognized as a crucial factor in neurodegenerative and neurological disorders. Moreover, human studies indicated the correlation between gut dysbiosis and major neurological and psychiatric disorders. The experimental results showed the importance of designing a novel microbiota-based probiotic dietary supplementation with the aim to prevent or ease the symptoms of AD, PD, or other forms of dementia. Promisingly, neuropsychiatric and neurodegenerative problems can be ameliorated through targeting the microbiota by microbiota transplantation, antibiotic treatment, or the administration of properly selected probiotics. The results of numerous studies indicate the beneficial effect of probiotic administration, including intestinal epithelial integrity enhancement and a protective role against neuroinflammation, neurodegeneration, and barrier disruption. Such solutions in the future may open new therapeutic paths and provide hope for improving the living condition of patients suffering from neurodegenerative diseases.

## Figures and Tables

**Figure 1 nutrients-14-03967-f001:**
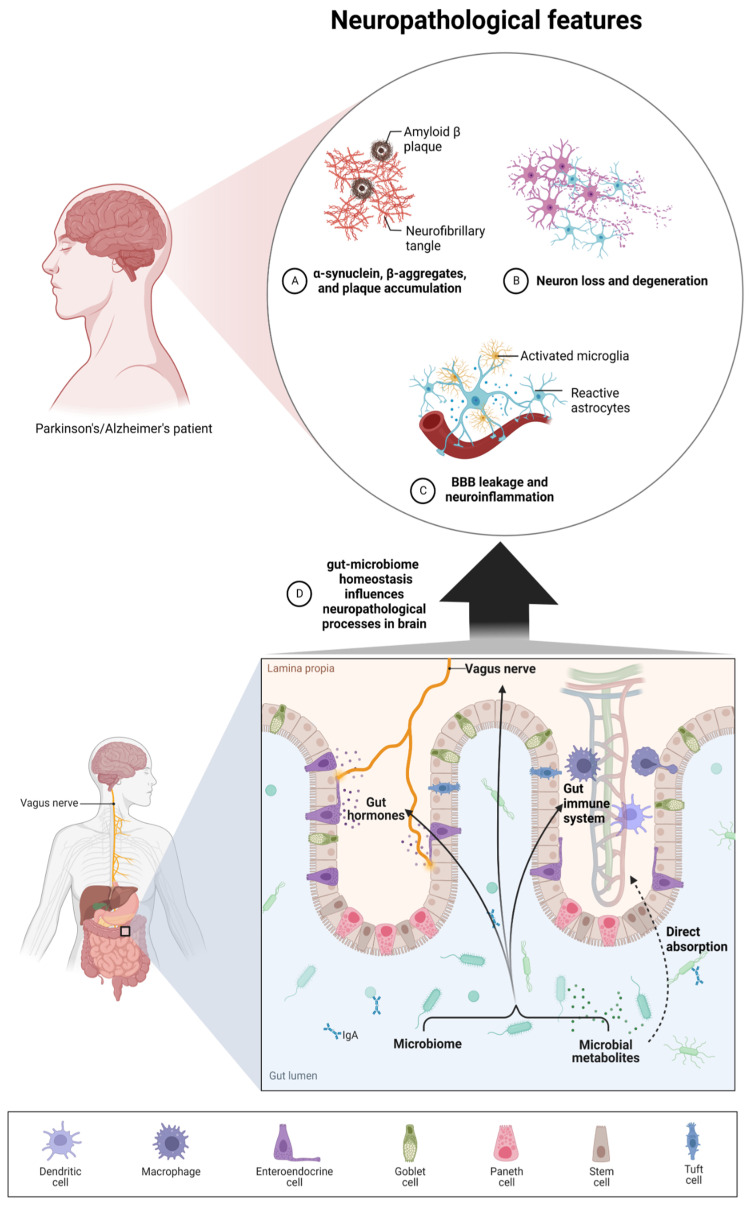
The interplay between the microbiome, microbial metabolites, and neuropathological processes present during neurodegenerative diseases such as AD and PD. The microbiome and microbial metabolites could regulate brain homeostasis via four different ways: 1. direct absorption of specific metabolites which have the ability to cross the BBB; 2. interaction between peripheral immune systems, which interact with brain glia cells and astrocytes; 3. influence on gut hormones and the enteroendocrine system; 4. communication via the vagus nerve. These four routes may have an influence on pathological symptoms such as increased BBB leakage and neural inflammation, enhanced beta-amyloid and alpha-synuclein formation, and finally neuron loss and motor/cognitive deficits.

**Figure 2 nutrients-14-03967-f002:**
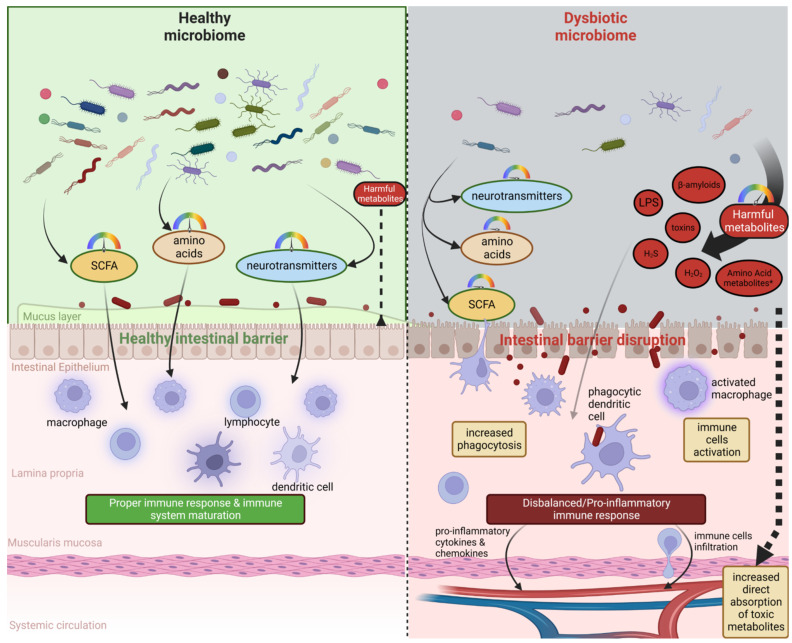
A healthy, abundant, and diverse microbiome (left side) is able to produce proper amounts of SCFAs, amino acids, and neurotransmitters. This type of microbiome and its metabolites positively influence gut barrier integrity and leads to the healthy maturation of the peripheral immune system, which is able to counteract external immune stimuli properly. A dysbiotic microbiome (right side) could be characterized by reduced diversity and an abundance of species able to produce positive metabolites and an increased abundance of species producing harmful metabolites such as LPS, beta-amyloids, small toxic metabolites, or other toxins. These switches may lead to disruption in the intestinal barrier, increased absorption of toxic metabolites, over-activation of the immune system, and, finally, pro-inflammatory responses which influence other organs, including the brain.

**Table 1 nutrients-14-03967-t001:** Changes in microbiome composition, its metabolic status, and influence on host homeostasis and metabolome in AD and PD.

Alzheimer’s Disease				
Microbiome Quantitative and Qualitative Changes	Changes in Microbiome Metabolome	Changes in Human Metabolome	Systemic Effects	Citation
*Eubacterium* deficiency; *Escherichia*, *Shigella* abundance	Increased LPS	Increased *β*-amyloid levels Increased plasma LPS levels	Increased inflammatory state	[74,93]
*H. pylori* infection	Increased levels of H_2_O_2_	Increased levels of homocysteine	Disrupted BBB integrity	[98]
Increased abundance of: *Escherichia coli*, *Bacillus subtilis*, *Klebsiella pneumoniae*, *Mycobacterium* and *Salmonella* species, *Staphylococcus aureus*, and *Streptococcus* spp.	Increased production of A*β*-amyloid plaques	Increased overall A*β*-amyloid plaques deposits	Disruption of proteostasis via molecular mimicry mechanism	[95]
Reduced: *Clostridium sporogenes* and *Ruminococcus gnavus*	Reduced levels of tryptamine	Reduced release of serotonin by enterochromaffin cells	Reduced levels of serotonin in gut	[48]
Not specified	Increased amount of secondary bile acids (deoxycholic acid, glycodeoxycholic acid, taurodeoxycholic acid or glycolithocholic acid)	Increased ratio between secondary bile acids and primary bile acids	Positive correlation between increased levels of secondary BAs, and hallmarks of AD	[91,100]
Parkinson’s disease				
Increased levels of *Akkermansia muciniphila* and *Bilophila wadsworthia*	Increased levels of S_2_^-^and SO_3_^-^ Increased degradation of taurine-conjugated bile acids	Disrupted transsulfuration pathways Reduced levels of glutathione	Increased oxidative stress peripheral and neuroinflammation	[135,139]
Reduced levels of *Prevotellaceae, Faecalibacterium*, and *Lachnospiraceae*	Reduced SCFAs production, increased proteolytic activity (production of *p*-cresol or phenylacetylglutamine)	Increased SCFAs concentration in plasma Possibly connected to lower levels of methionine in serum	Impaired gut-barrier function, increasing the risk of infection with enteric pathogens and boosting the *α*-synuclein formation	[119,122,123]

**Table 2 nutrients-14-03967-t002:** Clinical trials developed regarding probiotics administration and brain-related disorders.

Disease/Disorder	Study Title/Acronym	Administered Probiotic Strain Type	Observations	Government Identifier	Citation
Stress Anxiety Insomnia Depression	Psychiatric Symptoms in Employees Experiencing High Levels of Stress Before and After the Intake of Probiotics	(1) *Lactobacillus plantarum*: 10 billion CFU (colony forming units) (2) *Lactobacillus paracasei*: 10 billion CFU (3) The heat-treated *Lactobacillus paracasei*: 10 billion cells	Perceived stress, anxiety, and the improvement of related biological markers	NCT04452253	[215,216]
Emotional Stress	Probiotic Effects on the Microbe-brain-gut Interaction and Brain Activity During Stress Tasks in Healthy Subjects	*Lactobacillus helveticus*, *Bifidobacterium longum*, and *Lactiplantibacillus plantarum* in addition to other nutrients: 3 billion CFU per 3 g powder sachet	Improvement of brain function and emotional regulation	NCT03615651	[217]
Cognitive disorders Dementia	Probiotic on Psychological and Cognitive Effects	*Lactobacillus Rhamnosus* GG (L.GG)	L.GG did not lead to acute cognitive improvements for older adults already meeting physical activity guidelines	NCT03080818	[218]
Autism Spectrum Disorder	Probiotics and Oxytocin Nasal Spray on Social Behaviours of Autism Spectrum Disorder (ASD) Children	*Lactobacillus plantarum*: 200 million CFU per day	Social behavior improvement	NCT03337035	[219,220]
Cognitive Decline	The Cognitive Effects of 6 Weeks Administration With a Probiotic	*Lactobacillus paracasei* Lpc-37	Improvement of the cognitive and mood effects	NCT03601559	Not provided
Stress-related intestinal hyperpermeability	PRObiotic and Stress-related PERmeability (ProSPer)	A fresh fermented dairy drink containing *L. rhamnosus* CNCM I-3690 probiotic strain	*L. rhamnosus* CNCM I-3690 prevented stress-induced hyperpermeability to mannitol	NCT03408691	[221,222]
Mood Disorders Depression	Probiotics Therapy of Mood Disorders	Probiotic Ecologic^®^Barrier (Winclove Probiotics BV)	Mood improvement and the reduction of depressive symptoms in females in the perimenopausal age group	NCT04753944	Not provided
Epilepsy Stress Related Disorders	The Effect of Probiotic Supplementation in Drug-resistant Epilepsy Patients	*Streptococcus thermophilus*, *Lactobacillus acidophilus*, *L. plantarum*, *L. paracasei*, *L. delbrueckii* subsp. *bulgaricus*, *Bifidobacterium breve*, *B. longus*, *B. infantis*	Probiotic strains can alleviate stress-related disorders such as anxiety and depression	NCT03403907	[223,224]
Mood Disorders	Investigating a Probiotic on Mothers’ Mood and Stress (Promote)	*Bifidobacterium longum* (BL) NCC3001	Modulation of perinatal mood and stress during the perinatal period	NCT04685252	Not provided
Cognitive Impairment	Effect of Mediterranean Diet and Probiotics in Adults With Mild Cognitive Impairment	10^9^ colony forming units of *Lactobacillus rhamnosus* and *Bifidobacterium longum*	Cognitive change in Alzheimer’s Disease Assessment Scale-Cognitive-Plus (“ADAS-Cog-Plus”).	NCT05029765	Not provided
Major Depressive Disorder	Gut Feeling: Understanding the Mechanisms Underlying the Antidepressant Properties of Probiotics (PROMEX)	Multi-strain probiotic containing 14 strains	Alleviation of depressive symptoms in patients with Major Depressive Disorder (MDD)	NCT03893162	Not provided
Mild Traumatic Brain InjuryPost Traumatic Stress Disorder	Biological Signatures, Probiotic Among Those With mTBI and PTSD	*Lactobacillus reuteri* (*L. reuteri*; DSM 17938)	*L. reuteri* DSM 17938 exerted anti-inflammatory/immunoregulatory activity	NCT02723344	[225]
Depressive Symptoms Anxiety Stress	Probiotics, Brain Structure and Psychological Variables (ProBrain01)	Dietary Supplement: Vivomixx^®^ containing 8 probiotic strains	*Clostridium butyricum* regulates the gut microbiota and restores the butyrate content in the feces and the brains. *L. plantarum* PS128 daily intake improves anxiety-like behaviors and ameliorates neuropsychiatric disorders. Probiotic administration exerts positive results on all measures of depressive symptoms.	NCT03478527	[226,227,228,229,230,231,232,233]
Psychological Stress	Study on the Effects of a Probiotic on Autonomic and Psychological Stress	*Lactobacillus helveticus* R0052 and *Bifidobacterium longum* subsp. *longum* R0175	Stress alleviation, cortisol level reduction, positive effects on brain activity	NCT02417454	[234,235]
Severe Depression	Probiotic Supplementation in Severe Depression	Dietary Supplement: Vivomixx^®^ containing 8 probiotic strains	The increase of the *Lactobacillus* was associated with decreased depressive symptoms. Probiotic administration ameliorates depressive symptoms together with changes in the gut microbiota and brain.	NCT02957591	[236]
Acute Stress		Not specified: capsule containing freeze dried probiotic		NCT03284905	Not provided
Depression Sleep Disorders Stress Anxiety Mood Disorders	Probiotic Administration on Mood (PAM)	*Lactobacillus fermentum*, *Lactobacillus rhamnosus*, *Lactobacillus plantarum*, *Bifidobacterium longum*	Alleviation of the symptoms related to depression, anxiety, stress, insomnia, and emotional responses in healthy males and women.	NCT05343533	Not provided
Bipolar Disorder Schizoaffective Disorder	Probiotics to Prevent Relapse After Hospitalization for Mania	10^8^ CFU of the *Lactobacillus* GG and *Bifidobacterium lactis* strain Bb12.	Probiotic administration reduced the psychiatric rehospitalizations	NCT01731171	[237]
Impulsive Behaviour Compulsive Disorder, ADHD Borderline Personality Disorder	Treating Impulsivity in Adults With Probiotics (PROBIA)	*Pediococcus pentosaceus* 5–33:3, *Lactobacillus paracasei* subsp paracasei 19, and *Lactobacillus plantarum* 2362 in combination with four fermentable fibres	Ameliorating the impulsivity, compulsivity, and aggression in adults with psychiatric disorders	NCT03495375	[238]
Depression	Effect of Probiotic on Depression	*Lactobacillus paracasei*, *Bifidobacterium animalis*, *Bifidobacterium longum*, *Bifidobacterium bifidum* and *Lactobacillus plantarum*	Gut microbiota regulation and potential in alleviating depression	NCT04567147	[232,239,240,241]
Autism Spectrum Disorders, Anxiety	Probiotics for Quality of Life in Autism Spectrum Disorders	“Visbiome Extra Strength”-a mix of 8 strains of beneficial bacteria (mainly *Bifidobacteria* and *Lactobacilli*)	Probiotic administration improves GI and pain symptoms, also reducing anxiety and ASD-related behaviors	NCT02903030	Not provided
Schizophrenia Schizoaffective Disorder	Double-Blind Trial of a Probiotic Supplement to Reduce the Symptoms of Schizophrenia	*Lactobacillus rhamnosus* GG and *Bifidobacterium animalis* subsp. *lactis* (BB12)	Probiotic administration caused a beneficial change in Positive and Negative Syndrome Scale after probiotic administration and positive effect on GI tract	NCT01242371	[242]
Depression Anxiety	Effects of Probiotics on Mood	*Lactobacillus fermentum*, *Lactobacillus rhamnosus*, *Lactobacillus plantarum*, *Bifidobacterium longum*	Probiotic mixture as an adjuvant therapy for depression and anxiety. Probiotics administration has positive effects on depressive feelings.	NCT03539263	[243]
Anxiety	The Probiotic Study: Using Bacteria to Calm Your Mind	*Lactobacillus rhamnosus*	Anxiety and abdominal pain reduction after probiotic administration.	NCT02711800	[244]
Epilepsy	Probiotic in Treatment of Adult Patients With Drug-resistant Epilepsy	Combination of *Lactobacillus* and *Bifidobacterium* species	Not provided	NCT05160350	Not provided
Social Stress	Effect of Probiotics on Central Nervous System Functions in Humans	*Bifidobacterium longum* 1714	Probiotics can improve the response to social stress in healthy participants and patients with irritable bowel syndrome (IBS)	NCT02793193	[245]
Physiological Stress, Cognitive Decline	Effects of Probiotics on Cognition and Health (EPOCH)	Fermented milk (probiotic)	Fermented dairy consumption increased the presence of certain microorganisms in the gut and improved relational memory in healthy adults.	NCT02849275	[246]
Parkinson’s Disease	Trial of Probiotics for Constipation in Parkinson’s Disease	*Lactobacillus acidophilus*, *L. reuteri*, *L. gasseri*, *L. rhamnosus*, *Bifidobacterium bifidum*, *B. longum*, *Enterococcus faecalis*, *Enterococcus faecium*	Multi-strain probiotics treatment was effective for constipation in PD.	NCT03377322	[247]
Mental Fatigue Cognitive Function Mood Disorders	Examining the Effects of One-Month Probiotic Treatment on Mental Fatigue	A novel probiotic formulation (not specified)	Anti-fatigue effects of a probiotic supplement after a period of cognitive demand.	NCT03611478	Not provided
Autism Spectrum Disorder	Gut to Brain Interaction in Autism. Role of Probiotics on Clinical, Biochemical and Neurophysiological Parameters	Dietary Supplement: Vivomixx^®^	Vivomixx administration influenced inflammatory and gastrointestinal (GI) biomarkers, disturbances, behavioural and developmental profiles, and neurophysiological features in ASD pre-schoolers.	NCT02708901	[248,249,250]
Major Depressive Disorder Depression Depressive Symptoms	The Efficacy, Safety, and Tolerability of Probiotics on the Mood and Cognition of Depressed Patients	*Lactobacillus helveticus* and *Bifidobacterium longum*	Positive changes in mood, anxiety, cognition, and sleep after probiotic administration.	NCT0283804	Not provided
Psychological stress	A Study to Assess the Safety and Efficacy of Probiotic to Modulate Psychological Stress	Probiotic (not specified)	Probiotics can modulate the psychological stress experienced by healthy medical, dental and health science students.	NCT04125810	Not provided
Psychological Stress	Stress & Anxiety Dampening Effects of a Probiotic Supplement Compared to Placebo in Healthy Subjects	*Lacticaseibacillus paracasei* Lpc-37 at 1.75 × 10^10^ CFU per day	Probiotic administration can modulate stress and anxiety experienced by healthy subjects during and after an acute stressor compared to placebo.	NCT03494725	[251]
Anxiety Stress	Evaluation of a Probiotic On Anxiety and Stress in Healthy Adults Sensible to Daily Stress (BIOSTRESS)	PROBIOSTICK^®^	Probiotic administration decreases stress and anxiety of people sensible to daily stress.	NCT00807157	Not provided
Sleep Disorders	Nutritional Trial With Probiotic Fortified Milk in Women Affected by Insomnia (Prosit)	Fortified milk with *Bifidobacteria* infant-type and/or *Lactobacilli*	Functional milk administration improved sleep efficiency.	NCT03985228	Not provided
Autism Spectrum Disorder	Efficacy of Vivomixx on Behaviour and Gut Function in Autism Spectrum Disorder (VIVO-ASD)	Dietary Supplement: Vivomixx	Three-month supplementation with the Vivomixx improved overall function, aberrant behaviours, and the frequency of gastrointestinal symptoms in children with Autism Spectrum Disorders and co-morbid gastrointestinal symptoms.	NCT03369431	Not provided
Depression Anxiety Disorder	Effect of *Lactobacillus Plantarum* 299v Supplementation on Major Depression Treatment	*Lactobacillus Plantarum* 299V	*Lactobacillus Plantarum* administration improved the patient’s general health condition during antidepressant monotherapy with SSRI (Selective Serotonin Reuptake Inhibitor) in patients with major depression.	NCT02469545	Not provided
Autism	Effect of Milk Oligosaccharides and *Bifidobacteria* on the Intestinal Microflora of Children With Autism	Synbiotic containing *Bifidobacterium infantis* SC268, bovine colostrum and bovine oligosaccharides.	Synbiotic administration can promote a healthy bacterial environment in the intestines of children with autism spectrum disorders and gastrointestinal complaints.	NCT02086110	Not provided
Subjective Sleep Quality Objective Sleep Quality	Effect of *B. Longum* 1714™ on Sleep Quality	1 × 10^9^ CFU *Bifidobacterium longum* 1714™	Probiotic administration improved sleep quality	NCT04167475	Not provided
	Evaluation of efficacy of SAMEUP in subjects with depression symptoms: a randomized study (SAMEUP)	Combination Product: SAMEUp containing S-adenosylmethionine (SAMe) 200 mg and *Lactobacillus plantarum* HEAL9 1 × 10^9^ CFU	Improvement in the overall depression symptomatology after synbiotic administration.	NCT03932474	[252]

## Data Availability

Not applicable.

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
