# Peer review of "Does the Gut Microbial Metabolome Really Matter? The Connection between GUT Metabolome and Neurological Disorders"

_nutrients, 2022, doi:10.3390/nu14193967_

Round 1
Reviewer 1 Report
Marc et al. summarise the influence of microbial metabolites on severity, progression of neurodegenerative diseases and their correlation with microbial AD/PD-associated dysbiosis, pointing out particular genera/phyla responsible for increased/decreased production. Overall, this review is well organized and carries up-to-date findings.
A few minor concerns:
1. Acronyms should be used correctly. There are too many errors detected. Besides, many acronyms do not have full names at the first appearance.
2. In text cited references should be written correctly, such as [5, 6] instead of [5] [6].
3. To cite a work, may write Braak et al. instead of Braak H et al. (line 390).
4. The template is for Biomedicine. Please correct it.
5. Typos detected: line 454, AD i PD;
Author Response
Dear Reviewer 1,
Thank you for your help in improving the manuscript and all your comments. We have applied all suggested changes, which you can find in the corrected manuscript following track-change and text in "green" colour (suggestions of another 2 Reviewers).
Besides, herein you can find a short summary of introduced changes:
- Acronyms should be used correctly. There are too many errors detected. Besides, many acronyms do not have full names at the first appearance - Thank you for your valuable comment, we made important changes according your suggestions and updates in this issue.
- In text cited references should be written correctly, such as [5, 6] instead of [5] [6]- We have made corrections, which you can find in updated manuscript.
- To cite a work, may write Braak et al. instead of Braak H et al. (line 390). Thank you, we applied your suggestion.
- The template is for Biomedicine. Please correct it. Thank you - it has been corrected for Nutrients template.
- Typos detected: line 454, AD i PD; Thank you - corrected.
Kind regards,
Małgorzata Anna Marć
Reviewer 2 Report
The manuscript "Does the gut metabolome really matter?-Connection between gut metabolome and neurological disorders" sumarize the bibliography about the correlation between alterations of gut microbioma and brain-related diseases. The review is interesting and describes deeply the topic. However, it would be interesting and informative to include a table and a small description of Clinical Trials developed about microbiome and/or probiotics and these brain-related disorders (Alzheimer, Parkinson...)
Author Response
Dear Reviewer 2,
Thank you for your help in improving the manuscript and your valuable comments. We have inserted a new sub-chapter and a new table (marked in a "green" coulour), which contains a short desription of available and ended clinical trials regarding the probiotics administration and brain-related disorders/conditions (please see Table 2. in the updated manuscript). All the changes are available under track-change record in the corrected manuscript or are marked in green.
Kind regards,
Małgorzata Anna Marć
Reviewer 3 Report
This is a comprehensive review of the topic and the authors have done a great job covering it. The figures are particularly useful and review of the literature is more than sufficient. My only suggestion would be, where appropriate, to specify if male or female subjects were used in the studies cited. There are significant sex differences in many of the disorders covered, particularly Alzheimer's disease, and adding sex of study participants is something that should be encouraged in reviews of this sort.
Author Response
Dear Reviewer 3,
Thank you for your help in improving the manuscript and your valuable comments. We have inserted a new sub-chapter regarding "gender aspects" implications in neurodegenerative diseases. All the changes are available under track-change record in the corrected manuscript or are marked in green.
Kind regards,
Małgorzata Anna Marć